# The History, Efficacy, and Safety of Potential Therapeutics: A Narrative Overview of the Complex Life of COVID-19

**DOI:** 10.3390/ijerph18030955

**Published:** 2021-01-22

**Authors:** Farah Daou, Gretta Abou-Sleymane, Danielle A. Badro, Nagham Khanafer, Mansour Tobaiqy, Achraf Al Faraj

**Affiliations:** 1Faculty of Health Sciences, American University of Science and Technology, Beirut 1100, Lebanon; farah.daou92@gmail.com (F.D.); gabousleymane@aust.edu.lb (G.A.-S.); dbadro@aust.edu.lb (D.A.B.); 2INSPECT-LB, Institut National de Santé Publique, Epidémiologie Clinique et Toxicologie, Beirut 1100, Lebanon; 3Service d’Hygiène, Épidémiologie et Prévention, Hôpital Edouard Herriot, Hospices Civils de Lyon, 69003 Lyon, France; naghamkhanafer@hotmail.com; 4Centre International de Recherche en Infectiologie, Institut National de la Santé et de la Recherche Médicale U1111, Centre National de la Recherche Scientifique UMR5308, Ecole Normale Supérieure de Lyon, Université Claude Bernard 1, 69364 Lyon, France; 5College of Medicine, Department of Pharmacology, University of Jeddah, Jeddah 21959, Saudi Arabia

**Keywords:** COVID-19, safety, efficacy, therapeutics

## Abstract

The severe acute respiratory syndrome coronavirus 2 (SARS-CoV-2) pandemic posed a serious public health concern and started a race against time for researchers to discover an effective and safe therapy for coronavirus disease 2019 (COVID-19), the disease caused by SARS-CoV-2. This review aims to describe the history, efficacy, and safety of five potential therapeutics for COVID-19, remdesivir, favipiravir, hydroxychloroquine, tocilizumab, and convalescent plasma. A literature review was conducted through October 2020 to identify published studies evaluating the efficacy and safety of these five potential therapeutics. Clinical improvement was used to assess the efficacy, while reported withdrawals from study participation and adverse events were used to evaluate the safety. In total, 95 clinical studies (6 interventional and 89 observational studies) were obtained, of which 42 were included in this review. The evaluation of the efficacy and safety profiles is challenging due to the limitations of the clinical studies on one hand, and the limited number of randomized controlled trials (RCTs) on the other. Moreover, there was insufficient evidence to support repurposing remdesivir, favipiravir, and tocilizumab for COVID-19.

## 1. Introduction

Coronaviruses have been the cause of three outbreaks over the past two decades, starting with severe acute respiratory syndrome coronavirus (SARS-CoV) in 2002–2003 [1], Middle East respiratory syndrome coronavirus (MERS-CoV) in 2012 [2], and severe acute respiratory syndrome coronavirus 2 (SARS-CoV-2) in 2019–2020 [3]. Although these three pathogens are all members of the same family of viruses, the novel coronavirus has posed a more serious public health challenge due to its rapid and large-scale spread. The latter is due to three proposed factors: first, the characteristic transmissibility that contributed to more efficient human-to-human transmission; second, the characteristic pathogenicity that resulted in higher community transmission [4]; and third, the increased level of globalization that fueled the spread worldwide [5].

As of November 2020, the total number of cases and deaths caused by SARS-CoV-2 [6] already far exceed those caused by SARS-CoV [7] and MERS-CoV [8]. Preliminary assumptions also showed that SARS-CoV-2 has a lower case fatality rate (CFR) [9] in comparison to SARS-CoV [7] and MERS-CoV [8], and a higher basic reproductive number (R0) [9] in comparison to MERS-CoV [8], which further explain the rapid expansion of cases worldwide. It is crucial, however, to emphasize that the epidemiological features of the novel coronavirus are not yet well understood.

Another factor that magnified the situation was the absence of an approved vaccine or therapeutic agent until now, particularly as the severity of the coronavirus disease 2019 (COVID-19), the disease caused by the novel coronavirus, varies significantly and ranges between mild, moderate, severe, and critical illness [10]. As a result, clinical management is restricted to isolation, symptomatic treatment in mild and moderate illnesses, supportive management in severe and critical illnesses, and close monitoring of disease progression in all patients [11].

According to the World Health Organization (WHO), and as of 12 November 2020, there are currently 48 candidate vaccines in clinical evaluation and 164 candidate vaccines in preclinical evaluation [12]. On the other front, scientists are endeavoring to find therapeutics that can prevent, control, and treat COVID-19. Since the current situation poses an unprecedented challenge, clinical studies are underway to test the efficacy and safety of several repurposed therapies that showed efficacy against some coronavirus strains to treat COVID-19 [13] as a faster drug development pathway than traditional drug discovery. Repurposed therapies are approved drugs that include a list of antimalarial agents, antiretroviral agents, and antiviral agents, among other therapies; in addition to investigational drugs that include remdesivir and favipiravir. Moreover, adjunctive therapies are also being evaluated and these include corticosteroids, anticytokine or immunomodulatory agents, and immunoglobulin therapy [14].

This review aims to evaluate the efficacy and safety profiles of five agents proposed for the treatment of COVID-19: remdesivir, favipiravir, hydroxychloroquine, tocilizumab, and convalescent plasma.

## 2. Materials and Methods

A literature search of studies on SARS-CoV-2 was done in PubMed with the following combination of Medical Subject Heading terms: ((severe acute respiratory syndrome coronavirus 2) AND Humans[Mesh]) OR ((SARS-CoV-2) AND Humans[Mesh])) OR ((COVID-19) AND Humans[Mesh]) OR ((2019-nCoV) AND Humans[Mesh])), either solely or in combination with the names of the therapeutics. All English- and French-language observational and interventional studies published up to 1 October 2020 were included. Conference abstracts, review articles, and experimental studies were excluded.

Two authors (FD and AAF) screened article titles and abstracts in the initial search to identify those appropriate for inclusion. Subsequently, the full text of every article was read by each reviewer (GA, DB, NK, and MT). The results of the reviewers were compared and, in the case of disagreement, were resolved through discussion. In total, 95 clinical studies were obtained, out of which 42 were included in this review.

To assess the efficacy of the potential therapeutics in COVID-19 patients, clinical improvement and time to clinical improvement were assessed. To assess the safety of the potential therapeutics in COVID-19 patients, three parameters were evaluated:Withdrawals from study participation: defined as the percentage of patients who withdrew from the studies because of adverse drug events,Any adverse event: defined as the percentage of patients who reported adverse drug events of any grade,Serious adverse events: defined as the percentage of patients who reported serious adverse drug events (grade 3 or 4).

## 3. Results

The total number of studies retrieved after searching PubMed for studies evaluating the efficacy and safety of remdesivir, favipiravir, hydroxychloroquine, tocilizumab, and convalescent plasma in COVID-19 patients was 95 clinical studies (6 interventional and 89 observational studies), as presented in Figure 1, and of which 42 studies were included in this review.

### 3.1. Group A: Inhibitors of SARS-CoV-2 Replication

#### 3.1.1. Remdesivir

Remdesivir (GS-5734; Gilead Sciences Inc., Foster City, CA, USA) is a prodrug of a nucleotide analog, specifically an adenosine triphosphate (ATP) analog, that inhibits the function of viral RNA-dependent RNA polymerase (RdRp), and therefore, impedes the viral replication of a variety of RNA viruses [15].

It was first described in 2016 as a potential treatment for the Ebola virus (EBOV) after showing antiviral activity against EBOV in nonhuman primates [16]; however, based on an interim analysis by the safety monitoring board of the data from a clinical trial in 2019, the remdesivir arm was discontinued due to lower efficacy and higher mortality as opposed to the other investigational treatments [17]. In 2017, in vitro and preclinical in vivo studies demonstrated its activity against coronaviruses, including SARS-CoV and MERS-CoV, but no clinical trials were conducted to confirm this finding [18].

Remdesivir captured the headlines again in 2020 as a potential treatment for SARS-CoV-2. In vitro studies showed promising results [19], and although its mechanism of action is still under investigation, its activity against the RNA polymerase of SARS-CoV-2 was documented [20]. It was granted Emergency Use Authorization (EUA) in the United States by the Food and Drug Administration (FDA) on 1 May 2020 for hospitalized patients with severe COVID-19, which was expanded on 28 August 2020 to include hospitalized patients with suspected or laboratory-confirmed COVID-19 [21]. However, data from clinical studies conducted to date did not provide conclusive evidence on the efficacy of remdesivir in the treatment of COVID-19 patients. One case report suggested that remdesivir, unlike other antivirals that require immediate initiation after symptom onset, is effective in treating COVID-19 following late initiation [22], and two case series on the compassionate use of remdesivir in COVID-19 patients reported clinical improvement [23,24]. Randomized controlled trials (RCTs) on the other hand showed contradictory outcomes. Although the preliminary results of one RCT that included 1059 patients showed a faster time to recovery in the remdesivir group [25], another RCT on 237 patients reported that remdesivir did not reduce the time to recovery, or result in clinical improvement, or even accelerate viral clearance [26]. This lack of efficacy may be due to the fact that some viruses, including coronaviruses, can develop partial resistance to remdesivir as a result of mutations in RNA polymerase genes that trigger the enzyme’s proofreading activity, and this activity was reported in SARS-CoV-2 [15,27].

The safety profile of remdesivir has not yet been established [15], but almost all studies reported adverse events and withdrawals due to remdesivir administration. Withdrawals due to adverse events were reported and the percentages were 8% [24] and 22.8% [23] of participants in the case series, whereas in the RCTs, the percentages were as follows: 12% in the remdesivir group and 5% in the placebo group [26], versus 16% in the remdesivir group and 17% in the placebo group [25]. These percentages raise concerns about the tolerability of remdesivir.

Moreover, a relatively large number of participants receiving remdesivir developed adverse events of any grade. In one uncontrolled study, 60% of participants reported adverse events and 23% of participants showed serious adverse events with multiple-organ-dysfunction syndrome, septic shock, acute kidney injury, and hypotension being the most commonly observed [24]. In another uncontrolled study, four major adverse events were associated with remdesivir administration: hypertransaminasemia (42.8%), increased total bilirubin level (20.0%), acute kidney injury (22.8%), and rash (5.7%) [23].

Controlled studies revealed an alarming finding, since the rates of adverse events were overall similar between the remdesivir and placebo or control groups. In one controlled study, 66% of the remdesivir group and 64% of the control group reported adverse events, including constipation, hypoalbuminemia, hypokalemia, anemia, and increased total bilirubin, whereas serious adverse events were reported in 18% of the remdesivir group and 26% of the control group. The higher proportion of the control group recording serious adverse events may be because more patients in the remdesivir group (12%) than the placebo group (5%) discontinued treatment because of adverse events [26]. Similar findings were observed in another controlled study where 21.1% of the remdesivir group, compared to 27% of the control group, reported adverse events. The incidence of adverse events was not found to be significantly different between both groups, except for anemia, fever, hyperglycemia, and transaminase elevations being more common in the remdesivir group, and acute respiratory failure, hypotension, and viral pneumonia being more common in the control group [25].

In light of the results obtained by these clinical studies, and the increased media attention to remdesivir, the WHO recommended that remdesivir must not be administered as treatment or prophylaxis for COVID-19 outside of the context of clinical trials in its guide on the clinical management of COVID-19 that was issued on 27 May 2020 [11] (Figure 2a).

#### 3.1.2. Favipiravir

Favipiravir (Fujifilm Toyama Chemical Co. Ltd., Tokyo, Japan) is a modified pyrazine analog that inhibits RNA dependent RNA polymerase (RdRp), which ultimately prevents viral transcription and replication. It was originally developed and approved in Japan in 2014 to treat resistant cases of influenza [28]. The mechanism of action of favipiravir advocated it as a broad-spectrum antiviral against RNA viruses, and several in vitro and preclinical in vivo studies confirmed this, but clinical studies were limited to severe fever with thrombocytopenia syndrome (SFTS), Ebola virus infection, Lassa fever, and norovirus infection [29].

Due to its broad-spectrum antiviral activity against RNA viruses, favipiravir was suggested as a potential therapeutic against SARS-CoV-2, and its inhibitory action was demonstrated by an in vitro study [19]. However, data on the efficacy of favipiravir in COVID-19 patients are limited to the results of a few clinical studies. One observational study showed that favipiravir was superior to lopinavir/ritonavir combination therapy in accelerating viral clearance and improving chest computed tomography (CT) results [30]. On the other hand, an RCT (pre-print) demonstrated that favipiravir did not show superior efficacy to arbidol in terms of improving clinical recovery and was just associated with a significantly shorter duration to pyrexia and cough relief in comparison to arbidol [31]. However, favipiravir was not granted EUA in the United States by the FDA for patients with COVID-19.

The toxicity profile of favipiravir in humans is not yet established, and toxicity information is mainly based on animal studies. These studies showed that an overdose of favipiravir can be associated with vomiting, reduced body weight, and decreased locomotor activity. Repeated dose toxicity studies reported the following adverse events: testicular toxicity, increased vacuolization in hepatocytes, abnormal functioning of hematopoietic tissues such as decreased red blood cell (RBC) production, and elevated hepatic function parameters such as aspartate aminotransferase (AST), alkaline phosphatase (ALP), alanine aminotransferase (ALT), and total bilirubin. Moreover, favipiravir is classified as a teratogen and should not be used during pregnancy [32]. A recent review highlighted two other adverse events that were not reported in preclinical studies: hyperuricemia and QT prolongation [33].

In the observational study, favipiravir demonstrated a better safety profile, with 11.43% of participants reporting adverse events including diarrhea, liver injury, and poor diet compared to 55.56% in the lopinavir/ritonavir group [30]. However, the RCT (preprint) showed that arbidol was superior to favipiravir in terms of safety, since 23% of participants in the arbidol group reported adverse events compared to 32% in the favipiravir group, with abnormal hepatic function parameters, hyperuricemia, psychiatric symptom reactions, and digestive tract reactions reported in both groups [31]. Therefore, the efficacy and safety of favipiravir in COVID-19 patients cannot be confirmed; in particular, its superiority to other antivirals is inconclusive.

In addition to the insufficient evidence on the efficacy and safety of favipiravir in COVID-19 patients, its dose rationale initiated a debate within the scientific community. Some authors suggested that the 3200 mg loading dose on day 1 followed by the 1200 mg maintenance dose on day 2 to day 14 is effective [34] and safe [35] based on preliminary data. However, others expressed their concern that this dose rationale fails to achieve target drug concentrations based on data from clinical trials on EBOV and their own work on SARS-CoV-2 [36]. The latter was confirmed by one observational study which revealed that when this dosage was used, the trough concentrations of favipiravir in their participants with severe COVID-19 were much lower than those reported in healthy subjects [37].

In the middle of this uncertainty and the inclination of several governments to approve favipiravir for COVID-19, the WHO recommended that favipiravir must not be administered as treatment or prophylaxis for COVID-19 outside of the context of clinical trials in its guide on the clinical management of COVID-19 that was released on 27 May 2020 [11] (Figure 2b).

### 3.2. Group B: Inhibitors of SARS-CoV-2 Entry

#### Hydroxychloroquine

Hydroxychloroquine was originally used as an antimalarial agent, but its indications were later broadened due to its anti-inflammatory activity [38]. It was approved by the FDA for clinical use in 1955 [39]. Its exact mechanism of action is still unknown. It has been shown that it mainly targets lysosomes and interferes with antigen processing and presentation, as well as cytokine production. It is suggested that hydroxychloroquine inhibits viral entry, and the mechanism of action against SARS-CoV-2 and other similar viruses is two-fold: first, it increases the pH in endosomes and inhibits viruses from utilizing their activity for cell fusion and entry; second, it inhibits the glycosylation of angiotensin converting enzyme 2 (ACE2), which is the receptor targeted by some viruses, including SARS-CoV-2, for cell entry [38].

In vitro studies exhibited the potent antiviral activity of hydroxychloroquine against SARS-CoV-2 [40]. This was confirmed by published clinical data. In a non-randomized clinical trial, hydroxychloroquine (alone or coupled with azithromycin to prevent bacterial super-infection) resulted in viral load reduction within three to six days in COVID-19 patients [41]. More recent observational studies confirmed the previously reported efficacy of the combination of hydroxychloroquine and azithromycin in patients with COVID-19 [42,43]. Driven by the urgency of the current COVID-19 pandemic and the positive results shown by clinical studies, hydroxychloroquine was granted EUA in the United States by the FDA on 27 March 2020 for the treatment of COVID-19 in hospitalized adolescents and adults whose participation in clinical trials is not feasible [21]. However, four observational studies [44,45,46,47] and two RCTs [48,49], published soon after, were unable to confirm the efficacy of hydroxychloroquine, with or without azithromycin, in COVID-19 patients despite the severity of the disease. Therefore, data on the efficacy of hydroxychloroquine remain inconclusive, and further studies are needed [50].

Similar to other medications, hydroxychloroquine and azithromycin can cause side effects. Hydroxychloroquine is associated with headaches, drowsiness, visual disturbances, convulsions, and hypokalemia, in addition to cardiovascular adverse events, such as cardiovascular collapse, rhythm and conduction disorders including QT prolongation, torsades de pointes, ventricular tachycardia, and ventricular fibrillation, which may progress to sudden respiratory and cardiac arrest [38]. On the other hand, azithromycin is linked to hearing loss, hepatotoxicity, and renal toxicity, as well as cardiovascular adverse events including QT prolongation, cardiac arrhythmia, and torsades de pointes [51].

Only two studies reported non-serious adverse events, with one observational study reporting diarrhea, vomiting, nausea, and blurred vision [42] and one RCT reporting similar adverse events, in addition to fever, fatigue, and abnormal laboratory findings, among others [48]. Since cardiac arrhythmias are the most alarming side effects of hydroxychloroquine, electrocardiograms (ECG) changes were tracked in most studies. One observational study [43] and one RCT [48] did not report abnormal ECG findings among their participants, while one observational study discontinued hydroxychloroquine and azithromycin for participants who showed a QT interval above 500 ms, but did not provide information on the number of participants who were withdrawn [42]. However, abnormal ECG findings and cardiac arrest were reported in three observational studies [44,45,46] and one RCT [49]. Furthermore, one observational study [45] and one RCT [48] reported withdrawals.

Based on the evidence provided by larger clinical studies, the FDA issued a safety communication cautioning against the use of hydroxychloroquine for COVID-19 outside of a hospital setting or a clinical trial due to the risk of heart rhythm problems (arrhythmias) on 24 April 2020 [52], and on 15 June 2020, it revoked the EUA that was previously granted [53]. The WHO also recommended that hydroxychloroquine must not be administered as treatment or prophylaxis for COVID-19 outside of the context of clinical trials in its guide on the clinical management of COVID-19 that was issued on 27 May 2020 [11].

Another incident that added uncertainty to the already complicated situation involved the observational study published on 22 May 2020, which showed no clinical benefit for hydroxychloroquine or chloroquine with or without a macrolide in COVID-19 patients, and reported higher mortality and risk of arrhythmia in the treatment groups [46]. The journal released an “expression of concern” on 3 June 2020 [54], and “retracted” the article on 5 June 2020, because the data and analyses of the study could not be peer reviewed by a third-party after Surgisphere refused to transfer the required data, claiming that such a transfer would violate client agreements and confidentiality requirements [55] (Figure 3).

### 3.3. Group C: Adjunctive Therapies

#### 3.3.1. Tocilizumab

Tocilizumab is a monoclonal antibody (mAb) that targets the interlukin-6 (IL-6) receptor and blocks its activation [56]. It was first approved in Japan in 2005 for the treatment of Castleman’s disease [57], before gaining FDA approval for the treatment of some rheumatic diseases on 8 January 2010 [56]. In early March 2020, the existence of a “cytokine storm”, characterized by amplified cytokine release with multiorgan failure secondary to severe COVID-19, was put forward [58]. The involvement of IL-6 in the cytokine storm was also confirmed by a retrospective, multicenter cohort study that identified several risk factors for the mortality of adult inpatients with COVID-19 in Wuhan, China [59]. In view of these observational data, and the well-known central role of IL-6 in cytokine storms, tocilizumab was advocated as an adjunctive therapy in patients with severe COVID-19 [60,61]. The manufacturer received FDA approval on 18 March 2020 to conduct a randomized, double-blind, placebo-controlled phase III clinical trial to evaluate the safety and efficacy of intravenous (IV) tocilizumab as an adjunctive therapy in adult patients hospitalized with severe COVID-19 pneumonia [62]. However, tocilizumab was not granted EUA in the United States by the FDA for patients with COVID-19.

Several clinical studies involving COVID-19 patients described the positive outcome of tocilizumab administration with regards to attenuating the cytokine storm, relieving clinical symptoms, producing radiologic improvements, and halting progression to acute respiratory distress syndrome (ARDS) [63,64,65,66,67,68,69,70,71,72,73,74,75,76]. On the contrary, other clinical studies highlighted the lack of clinical usefulness of tocilizumab in COVID-19 patients [77,78,79,80]. However, there are no published results from RCTs at the time of writing of this manuscript, making the evaluation of the efficacy of tocilizumab as an adjunctive therapy for COVID-19 complicated.

Tocilizumab tolerance is generally good. The main adverse events include upper respiratory tract infections, nasopharyngitis, headaches, hypertension, and injection site reactions. However, other serious adverse events were also associated with tocilizumab administration, and these include serious infections (mainly if tocilizumab is administered during an active infection), gastrointestinal (GI) perforation, abnormal laboratory parameters (neutropenia, thrombocytopenia, and lipid abnormalities), immunosuppression, hypersensitivity reactions (including anaphylaxis), demyelinating disorders, and hepatotoxicity [81].

Some clinical studies reported adverse events following tocilizumab administration. In one observational study with 100 participants, three cases of severe adverse events were observed during the 10-day follow-up, where two participants developed septic shock and died, and one participant developed a gastrointestinal perforation requiring urgent surgery [71]. Additional findings were obtained from an observational study with 51 participants that reported the most common adverse events to be: hepatic enzyme elevations (29%), bacteremia (27%), thrombocytopenia (14%), neutropenia (6%) and cutaneous rash (2%) [79]. Another observational study with 25 participants showed that the majority of participants (92%) experienced at least one adverse event, the most frequent including anemia (64%), alanine aminotransferase (ALT) elevation (44%), and QT interval prolongation (20%) [63]. Elevated hepatic enzymes were reported in one case series as well [68].

The results of further clinical studies raised two concerns: first, an unusual adverse event was reported following tocilizumab administration in one patient with COVID-19 who developed viral myocarditis, suggesting that tocilizumab contribute to immunosuppression and promote viral replication if administered early in the disease course [80]; second, similar adverse events were reported in the tocilizumab and standard treatment groups in a case-control study with 65 participants, where serious adverse events were recorded in 25% of the tocilizumab group and 27% of the standard treatment group. Bacteremia, pulmonary thrombosis, pneumothorax, and hepatic enzyme elevations were the most common in both groups, and transitory neutropenia was only observed in the tocilizumab group [77].

Since the efficacy and safety of tocilizumab are still controversial, and there is not enough evidence to propose its use as an adjunctive therapy in patients with severe and critical COVID-19, the WHO recommended that tocilizumab must not be administered as treatment or prophylaxis for COVID-19 outside of the context of clinical trials in its guide on the clinical management of COVID-19 that was released on 27 May 2020 [11] (Figure 4a).

#### 3.3.2. Convalescent Plasma

Convalescent plasma transfusion was first discovered in the early 1890s by Behring and Kitasato, but was not used to treat a variety of infectious diseases until the 1930s. The concomitant emergence of antibiotics resulted in its short-lived use, as it was almost completely abandoned by the late 1940s. However, the interest in convalescent plasma transfusion has been renewed in recent years, and there is currently a number of immunoglobulin preparations that are either available to treat some infections, such as hepatitis B infection and rabies, or are in preclinical and clinical development [82].

During the current COVID-19 pandemic, convalescent plasma was considered a potential therapy for two reasons. First, some studies presented data on the ability of the immune system to mount an immune response to SARS-CoV-2 by detecting neutralizing immunoglobulin M (IgM) and immunoglobulin G (IgG) levels [83]. Second, SARS-CoV-2 was not shown to spread through the bloodborne route, making the risk of transmission via the convalescent plasma of recovered donors negligible [84].

Since no effective and safe therapy for COVID-19 has been approved to date, researchers exhibited an increased interest in convalescent plasma transfusions. The first published data were from a case series of five participants with critical COVID-19 illness who received convalescent plasma transfusions and showed improvements in clinical symptoms and viral clearance [85]. This drove the FDA to announce its efforts to develop and implement protocols for the use of convalescent plasma, as well as hyperimmune globulin that is derived from it, for the treatment of COVID-19 patients in the United States on 3 April 2020 [86]. Almost a month later, on 1 May 2020, the FDA approved the investigational use of convalescent plasma to treat critically ill patients and released “Recommendations for Investigational COVID-19 Convalescent Plasma” in an attempt to regulate convalescent plasma as an investigational product [87], and on 23 August 2020, it granted it EUA for the treatment of hospitalized patients with COVID-19 [21].

Recent published clinical studies showed positive outcomes in terms of clinical symptoms, radiologic changes, and viral clearance, thus emphasizing the adjunctive role of convalescent plasma in alleviating COVID-19 [88,89,90,91,92,93]. However, one controlled case series demonstrated that despite halting viral replication, convalescent plasma cannot reduce the mortality of COVID-19 patients with critical illness [94], and the only RCT on convalescence plasma demonstrated that it did not speed up the time to clinical improvement in patients with severe or critical COVID-19 illness [95].

Although data on the efficacy of convalescent plasma transfusions in COVID-19 patients do not reach a consensus on how effective it is, when to administer it in the course of the disease, how many infusions are needed, or what the optimal titers are to use, almost all clinical studies demonstrated that convalescent plasma was mainly associated with transfusion-related adverse events only. In one case series of 10 participants with severe COVID-19, only one patient showed an evanescent facial red spot [90]. Moreover, in the RCT which included 101 participants, only two reported transfusion-related adverse events, where one participant developed chills and rashes within 2 h of transfusion, and the other developed shortness of breath, cyanosis, and severe dyspnea within 6 h of transfusion, but both participants recovered fully after treatment [95]. However, the efficacy and safety of the use of convalescent plasma in COVID-19 patients requires more evidence to be properly assessed [96] (Figure 4b).

Figure 5 below highlights the body parts associated with adverse events related to the administration of the five potential therapeutics in COVID-19 patients.

## 4. Discussion

After being first used in 2003 by David Rothkopf during the SARS-CoV pandemic, the word “infodemic” gained public attention again in 2020 during the SARS-CoV-2 pandemic [97]. The remarkable information overload during the current pandemic left communities, scientists, and healthcare providers startled, and urged the WHO to initiate the “COVID-19 Myth Busters” section on its official website to provide reliable advice for the public [98]. However, scientists and healthcare providers are still overwhelmed with the information overload on one hand, and the conflicting evidence on the other. With 85,155 articles on PubMed [99] and 4344 clinical trials registered on ClinicalTrials.gov [100] up until January 2021, how can scientists and healthcare providers survive drowning in data relevant to COVID-19?

After reviewing the majority of clinical studies for remdesivir, favipiravir, hydroxychloroquine, tocilizumab and convalescent plasma in COVID-19 patients published through October 2020, and scrutinizing the efficacy and safety information presented by these studies, four major concerns were raised.

The primary concern is the unclear basis for the selection and prioritization of specific potential therapeutics for COVID-19 due to the scarcity of evidence on the efficacy profile, safety profile, or both. The therapeutics discussed in this review that cast doubts are remdesivir, favipiravir, and tocilizumab. Since remdesivir and favipiravir are investigational drugs, it is crucial to revise the results of completed clinical trials in registries to determine their suitability for repurposing them for the treatment of COVID-19. Therefore, ClinicalTrials.gov was checked and after excluding clinical trials for COVID-19, the following clinical trials were identified: for remdesivir, one phase II clinical trial for Ebola virus infection (NCT02818582) was completed, but no results were published, and one phase II clinical trial for detecting drug–drug interactions in patients with EBOV and human immunodeficiency virus (HIV) infections (NCT04385719) that is not recruiting yet. For favipiravir, three bioequivalence trials that are not yet recruiting, one pharmacokinetics study in patients with hepatic impairment that was completed but no results were published, five completed trials for influenza with the results of only one phase II trial (NCT01068912) published on the registry, and three trials for Ebola virus infection with one phase II trial (NCT02739477) terminated due to the end of the pandemic, one completed trial (NCT02662855) with unpublished results, and one completed trial (NCT02329054) suggesting that the efficacy and safety of favipiravir in Ebola virus infection is not firm. The latter suggests that experience with remdesivir and favipiravir in previous epidemics (mainly the EBOV, SARS-CoV, and MERS-CoV pandemics) does not establish conclusive evidence to support their repurposing for COVID-19, in contrary to what is postulated by some researchers. Tocilizumab, on the other hand, targets a single component of the cytokine storm associated with COVID-19 (IL-6), and although studies confirmed the involvement of IL-6, the clinical benefits of targeting IL-6 to mitigate the cytokine storm, as well as of immunosuppression in general in COVID-19 patients, is not well understood [101].

The secondary concern is the absence of safety endpoints in certain clinical studies, where no data on withdrawals and/or adverse events were presented. Studies that did not disclose such information are reported in the Appendix A. The reason why this observation is a cause for concern is the possibility that the absence of data does not necessarily indicate the absence of adverse events and/or withdrawals.

The tertiary concern is the methodological downsides of the clinical studies that hinder the evaluation of both efficacy and safety of these potential therapeutics, and cause the quality of evidence in support of or against these therapeutics to be sub-optimal. The methodological downsides of the clinical studies include the concomitant administration of other treatments, possible drug–drug interactions, elevated cumulative doses of the tested therapeutic, prolonged treatment duration, and administration at various stages of the disease course, in addition to the absence of control groups, lack of blinding, and discrepancies in the clinical endpoints. Another major limitation of some clinical studies is the disregard of the presence of comorbid conditions that may alter disease progression and response to these therapies [102]. It is essential to point out that some controlled studies did not disclose data on the standard of care (SOC) that served as a control arm as well.

The quaternary concern is the direct and indirect sponsorship of clinical trials for COVID-19 by the pharmaceutical industry. It is undeniable that this type of sponsorship can result in major advancements in therapeutics, but the downsides may outweigh this advantage. These downsides were revealed by several studies, and mainly include conflict of interest, methodological bias, reporting bias, publication bias, selecting an inappropriate comparator, and the concealment of adverse reactions. The latter cause industry-funded clinical trials to yield positive results much more frequently than studies funded by other entities [103]. Another main problem that emerged during the COVID-19 pandemic is the premature release of data from clinical studies sponsored by the pharmaceutical industry that was justified by the urgency of the situation, which can lead to an improper assessment of the efficacy and safety of the tested therapeutic.

This study has a number of limitations. First, not all potential agents to treat COVID-19 were included. Second, not all studies were evaluated, because of the sheer quantity of studies. We chose to analyze studies with select criteria because of the increased statistical reliability; yet, this approach may have biased the results. However, to the best of our knowledge, this is the first review that detailed the clinical history, efficacy, and safety of the five reviewed agents.

## 5. Conclusions

In this review, the focus was on five potential therapeutics for COVID-19 that gained publicity: remdesivir, favipiravir, hydroxychloroquine, tocilizumab, and convalescent plasma. Despite the availability of a relatively considerable amount of data, evidence on the efficacy and safety of these potential therapeutics is inconclusive because the data were mainly generated by observational studies, and limitations of the majority of the clinical studies caused the quality of evidence in support of or against these therapeutics to be sub-optimal.

Since the results of only a few RCTs have been published to date, and scientists are currently evaluating commercialized and new therapies [104,105], as well as working on new technologies (such as theranostic nanoparticles to improve drug delivery in COVID-19 patients [106]), it is essential to monitor the results of future clinical trials and to report all data, even if they show a lack of efficacy, higher frequency and severity of adverse events, or both.

## Figures and Tables

**Figure 1 ijerph-18-00955-f001:**
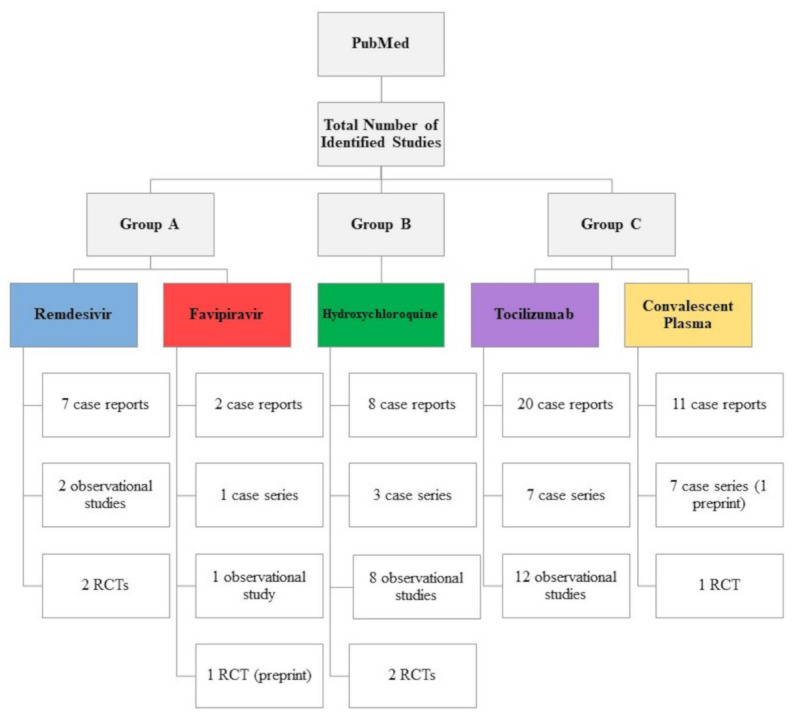
Number of studies retrieved for remdesivir, favipiravir, hydroxychloroquine, tocilizumab, and convalescent plasma in coronavirus disease 2019 (COVID-19) patients as of 1 October 2020. RCT: randomized controlled trial.

**Figure 2 ijerph-18-00955-f002:**
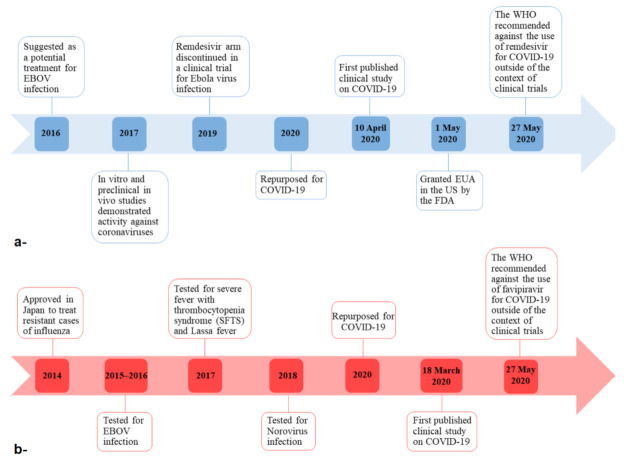
Timelines of remdesivir (**a**) and favipiravir (**b**) from discovery to repurposing for COVID-19. EBOV: Ebola virus; WHO: World Health Organization; FDA: Food and Drug Administration; EUA: Emergency Use Authorization.

**Figure 3 ijerph-18-00955-f003:**
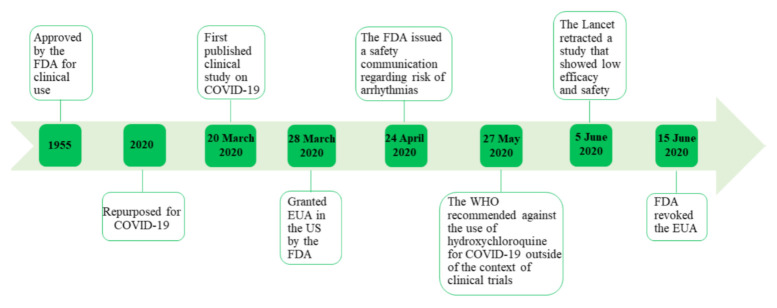
Timeline of hydroxychloroquine from FDA approval to repurposing for COVID-19.

**Figure 4 ijerph-18-00955-f004:**
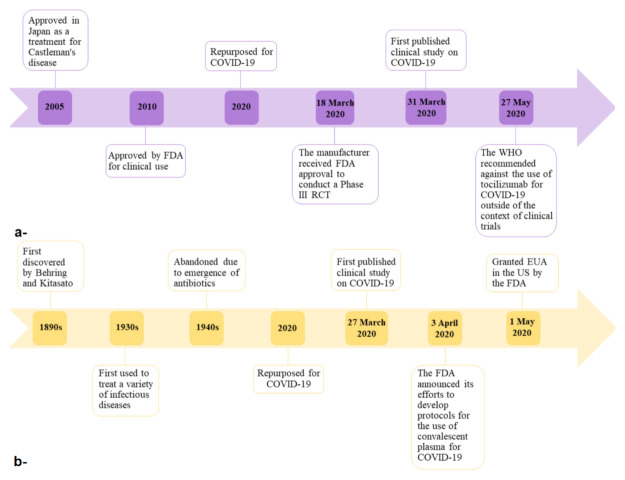
Timelines of tocilizumab (**a**) and convalescent plasma (**b**) from discovery to repurposing for COVID-19.

**Figure 5 ijerph-18-00955-f005:**
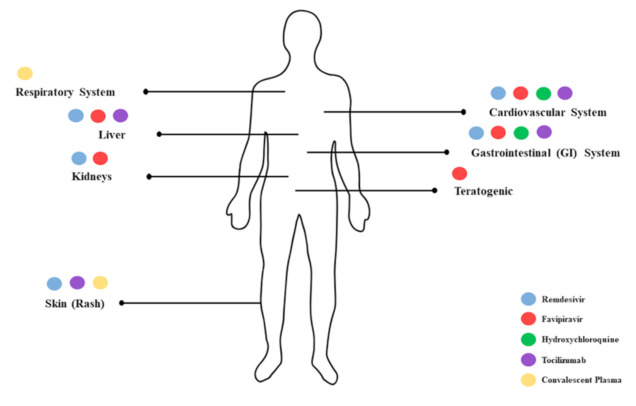
Summary of adverse events associated with the administration of the five potential therapeutics in COVID-19 patients.

## Data Availability

Not applicable.

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
