# Peer review of "The History, Efficacy, and Safety of Potential Therapeutics: A Narrative Overview of the Complex Life of COVID-19"

_ijerph, 2021, doi:10.3390/ijerph18030955_

Round 1

Reviewer 1 Report

This review by Daou and colleagues present an overview of the efficacy and safety profiles of five therapeutic agents proposed for the treatment of COVID-19. Although this is an interesting update on the state of the therapeutics, the article needs improvement prior to further consideration. Here are some suggestions that the authors may consider;

  1. The length of each section needs to substantially reduced with only key points discussed. Suggest corresponding to each drug, the key findings, clinical characteristics, adverse events and funding bias be put together in a Table. 
  2. A Cochrane review concluded that the evidence of convalescent plasma from people who have recovered from COVID‐19 being an effective treatment for COVID‐19 hospitalised patients is uncertain. Moreover, the evidence was low or very low. See https://doi.org/10.1002/14651858.CD013600.pub3
  3. Lines 156-160: Authors state that, "In one controlled study, 66% of 158 participants in the remdesivir group and 64% of participants in the control group reported adverse 159 events, and in both groups, constipation, hypoalbuminemia, hypokalemia, anemia, and increased 160 total bilirubin were the most commonly observed adverse events." Please cite this study. 
  4. Suggest rather than talking about individual studies, please discuss the main findings and overall evidence/criticism of the safety and efficacy of the drugs. Individual studies, adverse events, and specific indications could be summarised in a Table instead.
  5. In the context of Favipiravir, several studies discussed by the authors have varying comparator/control arms - which make it difficult to draw relevant conclusions. 
  6. With regards to hydroxychloroquine, authors should summarise main points concerning the adverse events and indications for COVID-19 patients in the context of the clinical trial or otherwise.
  7. In the "Conclusion", please summarise the main findings of the review at the outset.
  8. Authors should clearly specify that the quality of evidence in support or against some of these drugs is low or sub-optimal.
  9. Please provide a summary table of all adverse events reported corresponding to each agent discussed in this review. 
  10. The role of comorbidities in COVID-19 associated mortality and how they mediate the effects of therapy should also be discussed.

Reviewer 2 Report

1. The systematic review needs to adhere to Preferred Reporting Items for Systematic Reviews and Meta-Analyses: The PRISMA. The authors’ description of the methodology is very close to PRISMA and it wouldn’t take a large effort to conform to PRISMA. Please follow the steps outlined here:

The PRISMA statement for reporting systematic reviews and meta-analyses of studies that evaluate health care interventions: explanation and elaboration.
Liberati A, Altman DG, Tetzlaff J, Mulrow C, Gøtzsche PC, Ioannidis JP, Clarke M, Devereaux PJ, Kleijnen J, Moher D.PLoS Med. 2009 Jul 21;6(7):e1000100. doi: 10.1371/journal.pmed.1000100.

Preferred reporting items for systematic reviews and meta-analyses: the PRISMA statement.
Moher D, Liberati A, Tetzlaff J, Altman DG; PRISMA Group.PLoS Med. 2009 Jul 21;6(7):e1000097. doi: 10.1371/journal.pmed.1000097. Epub 2009 Jul 21.

2. Include the following study to the ones that did show hydroxychloroquine’s efficacy against COVID-19:

Treatment with hydroxychloroquine, azithromycin, and combination in patients hospitalized with COVID-19.
Arshad S, Kilgore P, Chaudhry ZS, Jacobsen G, Wang DD, Huitsing K, Brar I, Alangaden GJ, Ramesh MS, McKinnon JE, O'Neill W, Zervos M; Henry Ford COVID-19 Task Force.Int J Infect Dis. 2020 Aug;97:396-403. doi: 10.1016/j.ijid.2020.06.099. Epub 2020 Jul 2.

3. Include the latest RCT showing the lack of hydroxychloroquine’s efficacy against COVID-19:

Effect of Hydroxychloroquine on Clinical Status at 14 Days in Hospitalized Patients With COVID-19: A Randomized Clinical Trial.
Self WH et al
JAMA. 2020 Dec 1;324(21):2165-2176. doi: 10.1001/jama.2020.22240.

Reviewer 3 Report

The aim of the review submitted by Daou et al. was to identify and describe published studies evaluating the efficacy and safety of five potential therapeutics for COVID-19. The review is clear and very interesting; the paper is well-written. However, I have some points that should be revised.

I am sorry, but I did not understand the reason why you included in the review 52 clinical studies, instead of 93 that you identified. Please, could you better explain?

Minor revisions include:

  • Line 145: I suggest to change the word “ while” with “whereas”
  • Lines 199-200 ; 247-248 and 301-302: references should be added
  • Line 181-183: references should be added
  • Line 233: I suggest to change “dosage is used”, with “dosage was used”
  • Line 299: “ Group C: Adjunctive Therapies” should be written in italics
  • In “References”, some Journal name are written with the abbreviated form, whereas others are not. Please check throughout the text and modify the wrong form with the correct one (Journal name should be abbreviated).

Round 2

Reviewer 1 Report

The authors have partially addressed the concerns/comments provided. The article could be accepted should the concerns from other reviewers be addressed too.

This manuscript is a resubmission of an earlier submission. The following is a list of the peer review reports and author responses from that submission.